# Lessons from a 3-Year Review of PSMA PET-CT in a Tertiary Setting: Can We Fine Tune Referral Criteria by Identifying Factors Predicting Positivity and Negativity?

**DOI:** 10.3390/diagnostics13152542

**Published:** 2023-07-31

**Authors:** Vineet Pant, Sobhan Vinjamuri, Ahmad Zaid Zanial, Faisal Naeem

**Affiliations:** 1Department of Nuclear Medicine, Royal Liverpool University Hospital, Liverpool L78XP, UK; sobhan.vinjamuri@gmail.com (S.V.); faisal.naeem@liverpoolft.nhs.uk (F.N.); 2Department of Nuclear Medicine, General Hospital, Kuala Lumpur 50586, Malaysia; ahmadzaidx@gmail.com

**Keywords:** biochemical recurrence, prostate cancer, recurrence, indeterminate PSMA PET

## Abstract

Aim of the study: To draw inferences from a retrospective evaluation of PSMA PET CT scans performed for the evaluation of biochemical recurrence. Material and Methods: A retrospective analysis of 295 PSMA PET CT scans spanning 3 years between 2020 and 2022 was undertaken. Results: Of 295 PET CT scans, 179 were positive, 66 were negative and 50 had indeterminate findings. In the positive group, 67 had radical prostatectomy and PSMA avid lesions were seen most commonly in pelvic lymph nodes. The remaining 112 positive scans were in the non-radical prostatectomy group; 25 had recurrence only in the prostate, 17 had recurrence involving the prostate bed; 28 had no recurrence in the prostate gland, while 42 had recurrence in the prostate as well as in extra-prostatic sites. Overall, in the non-prostatectomy group, 75% of the population was harboring a PSMA avid lesion in the prostate gland while in the remaining 25% of the population, recurrence did not involve the prostate gland. The majority of indeterminate findings were seen in small pelvic or retroperitoneal lymph nodes or skeletal regions (ribs/others) and in nine patients indeterminate focus was seen in the prostate bed only. Follow-up PSMA PET CT was helpful in prior indeterminate findings and unexplained PSA rise. Conclusion: A higher recurrence in the prostate bed while evaluating biochemical recurrence prompts the following: question: should prostatectomy be offered more proactively? Follow-up PSMA PET CT is helpful for indeterminate findings; a PSA rise of 0.7 ng/mL in 6 months can result in positive PSMA PET CT while negative scans can be seen up to a 2 ng/mL PSA rise in 6 months.

## 1. Introduction

Prostate cancer (PCa) is the most common cancer in men in the Western world [1]. Despite definitive treatments in the forms of either radical prostatectomy (RP) or radiation, many men later develop a recurrence of prostate-specific antigen (PSA) with no evidence of disease on conventional imaging; this is called a biochemical recurrence (BCR) [2]. In the literature, up to 53% of men with prostate cancer have a biochemical recurrence of PSA after radical treatments [3].

Although an increase in PSA levels can indicate the progression of PCa, it cannot localize the clinical recurrence. Early detection of a recurrent disease provides the possibility of treatment with curative intent. Morphological imaging modalities (e.g., computed tomography (CT) and magnetic resonance imaging (MRI) have limited value in the detection of PCa lesions (metastases or recurrences) due to their low sensitivity and specificity [4].

Recently, PSMA-PET imaging is replacing conventional imaging for the evaluation of biochemically recurrent PCa based on its superior sensitivity and specificity. At present, several PSMA tracers are available for clinical use, including tracers labelled with ^68^Ga or ^18^F. ^18^F labelled tracers are considered more beneficial as compared to ^68^Ga, especially due to the lower kinetic properties of emitted positron and longer half-life. Nevertheless, high detection rates for recurrent lesions are documented for both ^68^Ga and ^18^F labelled tracers [5]. As a result, PET imaging with PSMA tracers for prostate cancer has found its way into standard clinical practice and is already incorporated in European and national guidelines.

However, the studies till now have focused on the diagnostic performance of PSMA PET CT and not their effect on care pathways. There are no trials indicating that early detection of recurrence by PSMA PET CT has changed the management and improved patient care [6].

Ours is the first evaluation of PSMA PET CT scans’ impact on clinical management and their implications which can lead to possible changes in referral criteria.

## 2. Material and Methods

In our study, we initially considered 333 18F PSMA 1007 PET CT scans conducted between 2020 and 2022, focusing specifically on evaluating biochemical recurrence. We excluded 38 scans that were used for primary staging, resulting in a final retrospective analysis of a subset of 295 scans.

To ensure a comprehensive analysis, we meticulously gathered clinical information, including age, Gleason score, histopathology, serum PSA levels at the time of scan request, and treatment history from available records. Our evaluation of the scans involved a thorough examination of local disease, lymph node involvement, and distant metastases, aiming to determine their potential impact on clinical management decisions.

Before undergoing PSMA PET CT scans for biochemical recurrence evaluation, all patients had previously undergone an MRI of the pelvis and CT of the abdomen/pelvis within three months of the PSMA PET CT scan. The PSMA scans were conducted within a timeframe of 20–68 days from the initial request. The slight variation in this interval was primarily influenced by regional factors, such as demographic characteristics, availability of radiopharmaceuticals, manpower resources, and patient preferences. Notably, only a limited number of patients referred for repeat PSMA PET CT had available PSA kinetics data for analysis.

### 2.1. Imaging

PET/CT studies were performed on Discovery 690 PET/CT systems (GE Healthcare, Technologies Inc, Chicago, IL, US). Patients were hydrated, asked to void immediately before the acquisition, and scanned from vertex to mid-thigh approximately 120 min after intravenously administered activity of 18F-PSMA-1007 as 4 MBq per kilogram bodyweight. PET acquisition was undertaken in the time-of-flight mode. A concurrent CT scan was acquired using automatic mA-modulation, and 120 kV. The images were reconstructed using the iterative reconstruction technique with attenuation correction provided by CT data. No intravenous or oral contrast was administered.

### 2.2. Image Evaluation

18F PSMA 1007 PET CT studies were reviewed by two experienced Nuclear Medicine physicians with more than 10 years of experience. Prior imaging in the form of CT, MRI or bone scan was also reviewed. Lesions were categorized using PSMA-RADS version 1.0 [7]. The scan findings were classified as positive for recurrence/metastasis (PSMA RADS 4/5), indeterminate (PSMA RADS 3) and negative (PSMA RADS 1/2).

Relevant equivocal lesions were considered for further evaluation by repeat PSMA PET CT after interdisciplinary review. None of the equivocal lesions (PSMA RADS 3 group) were doubtful for a non-prostate malignancy. It is likely in this group of patient population, as all these patients had earlier been evaluated with other cross-sectional imaging (MRI/CT) earlier, during the time of primary diagnosis of prostate malignancy and subsequently during biochemical relapse prior to the PSMA PET CT scan.

### 2.3. Statistical Analysis

Data were entered into the Excel sheet (Microsoft Excel) manually and statistical analyses were performed using Excel, confidence intervals were calculated and *p* values less than 0.05 were considered significant.

## 3. Results

These 295 PSMA PET CTs were done for 229 patients; 57 patients had PSMA PET CT done twice, three had it thrice and one patient had it four times; due to various reasons (indeterminate findings on a prior scan or further rise in PSA levels following prior negative PSMA scan and imaging with MRI, CT). Of the 295 PET CT scans, 179 were considered positive, 66 were negative and 50 had indeterminate findings (Figure 1).

In these scans being evaluated for BCR, the most common T stage; either post-operatively in the case of radical prostatectomy or radiologically was T3 (Table 1) among all groups. Among the positive group of 179 scans, PSA range at the time of scan request was 0.18–99.7 ng/mL and the most common Gleason’s score was seven; 67 (37.3%) scans had radical prostatectomy status (RP) and the rest received radiotherapy. Among the negative and indeterminate groups, the PSA range was understandably lower than the positive group with the PSA range of 0.08–3.5 ng/mL among the negative group and 0.14 to 6.5 among the indeterminate group. In the negative group, 92.4% had RP status, five received only radiotherapy while in 22 scans, patients had received additional salvage radiotherapy to the prostate bed and pelvic nodes after RP. In the indeterminate group, 28 had RP status, 22 received only radiotherapy, and 12 received salvage radiotherapy following RP.

In 179 positive scans, the highest number of positive findings were seen in lymph nodes with pelvic lymph nodes (112/179 scans) being most involved; 35 only had pelvic lymph nodes, six had only pelvic and retroperitoneal lymph nodes, and in the rest, pelvic lymph node were involved in various combinations along with the involvement of the prostate bed, bones and soft tissue. Among this positive PET CT group (Table 2); 67 had radical prostatectomy and of these PSMA avid lesions were seen most in pelvic lymph nodes and seen only in pelvic lymph nodes in 33, prostate bed and pelvic lymph nodes in four, pelvic and retroperitoneal lymph nodes along with bones in 13, only bone involvement in one and only soft tissue involvement (lungs) followed by pelvic, retroperitoneal lymph nodes along with bones (five scans), prostate bed along with pelvic nodes (four scans), only bones (two scans) and bone with soft tissue involvement (one scan). In 112 positive PET CT scans, in the non-radical prostatectomy group, 25 had recurrence only in the prostate, 17 had recurrence involving prostate and seminal vesicles (i.e., prostate bed) only; 28 had no recurrence in the prostate gland and only pelvic lymph nodal (two scan), pelvic and retroperitoneal lymph nodal (six scans) or systemic recurrence were present (20 scans, bone, visceral metastases); while 42 patients had a recurrence in the prostate as well as extraprostatic sites. Overall, in this non-prostatectomy recurrence group, in 75% of the population (84/112 scans) prostate glands was found to harbor a PSMA avid lesion while in 25% of the population recurrence did not involve the prostate gland.

In 50 PET CT scans with indeterminate findings, the majority (41/50 scans) of indeterminate findings were seen in small pelvic or retroperitoneal lymph nodes or skeletal regions (ribs/others) and only in nine patients indeterminate focus was seen in the prostate bed only.

In our analysis, we examined the scan findings in relation to the PSA levels at the time of the scan request. We observed that PSA levels ≥ 0.5 ng/mL were associated with a higher number of PSMA avid lesions (both positive or indeterminate uptake) on the scan in 184 out of 229 cases. Conversely, PSA levels < 0.5 ng/mL were more frequently associated with negative scan results in 48 out of 66 cases. However, it is important to note that these observations did not reach statistical significance, which is probably due to the small number in each subgroup. We also found similar trends when comparing PSA levels ≥ 1 ng/mL to PSA levels < 1 ng/mL (*p*-value 0.05, see Table 3).

In a separate analysis of patients having prostate gland at the time of scan (i.e., non-radical prostatectomy group), we found that in patients having prostate gland, PSMA PET CT showed a tracer avid lesion (positive or indeterminate) if PSA level ≥ 0.71. Further, in this group the majority were having tracer avid focus in the prostate gland either alone or in various combinations of the seminal vesicle, lymph nodal and systemic involvements.

We conducted a separate subgroup analysis of 61 patients (Table 4) who had repeat imaging. Thirty had indeterminate findings on initial PSMA PET CT and the findings changed to a positive lesion in 14 (10 with known small indeterminate pelvic lymph node and four with known indeterminate prostate focus, PSA increase of 0.7 to 3.0 in 6 months), negative in eight (four with pelvic lymph node, four with uptake in bones on prior PET CT, PSA increase of 0.13–2). Six patient had the same indeterminate small pelvic lymph node, two had the same indeterminate retroperitoneal lymph node (PSA increase of 0.4 to 2.5). Overall, the follow-up PSMA PET CT was able to conclude in 24/30 patients on follow-up. Of the 31 patients who had repeat PSMA PET CT scans done for a further rise in PSA levels (PSA rise range 0.3–10), 24 had new lesions (prostate lesion in four, prostate and seminal vesicle involvement in four, pelvic/retroperitoneal lymph nodes in 12, new bone lesion in four) and eight had negative scans even on repeat imaging (PSA rise 0.3–2.3). For example, patient in Figure 2 had negative scan at PSA of 3.4 ng/mL, PSA velocity of 1.3 ng/mL in six months. The uptake in right humerus is at the site of known prior humerus fracture and therefore it was categorized as PSMA RADS 1 B (7). Another patient in Figure 3 had and indeterminate lymph node at PSA 6.5 ng/mL, after rise of 3 ng/mL in 2 years, it was categorized as RADS 3A(7). An interesting finding in this case was the consolidation changes in the right upper lobe, which resolved in a follow-up CT performed after 2 months (Figure 4).

## 4. Discussion

PSMA-PET CT is being increasingly used in clinical practice and is slowly becoming indispensable in the care of many PCa patients by overcoming the challenges of low sensitivity and specificity of conventional imaging modalities. Studies focusing on diagnostic performances of PSMA PET CT in BCR have reported high detection rates (75–81%) in detecting PSMA-positive lesions. Xing Zhoe et al., while evaluating BCR in patients following RP, found a high detection rate for patients after radical prostatectomy. In their study, 79% of patients showed at least one pathological finding on ^18^F-PSMA-1007 PET/CT [8]. Treglia G et al. performed a systematic review and meta-analysis for the detection rate of ^18^F-labeled PSMA PET/CT in BCR of PCa and found a pooled diagnostic rate of 81% for ^18^F-labeled PSMA PET/CT [9]. Similarly, in our study, we were able to detect PSMA avid lesions in 229/295 scans (77%) which included both positive and indeterminate lesions as per PSMA RADS criteria.

In our study, the mean PSA levels in the positive PSMA group were higher than the indeterminate group and negative group, indicating higher detection rates at high PSA levels. The available literature also suggests higher detection rates at higher PSA levels. Xing Zhoe et al. reported higher detection rates in their population group with rising PSA levels, 50% detection rates at PSA levels ≤ 0.5 ng/mL and 10% at >2.0 ng/mL. Similarly, the pooled diagnostic rate for detecting recurrent lesions in systemic review by Treglia G et al. for 18 FSMA PET CT was 86% for PSA ≥ 0.5 ng/mL (95% CI: 78–93%) and 49% for PSA < 0.5 ng/mL (95% CI: 23–74%). Geisel et al. in their studies for BCR of PCa found a good diagnostic rate of 18 F-PSMA-1007 PET CT and also found it to be related to serum PSA values. They concluded that higher PSA values were associated with higher diagnostic rate of 18 F PSMA PET CT [10,11].

A few articles have evaluated the detection rate of 18 F PSMA PET CT at different serum PSA levels. A metanalysis by Ferrari et al. showed that the pooled diagnostic rate was 51% (95% CI: 29–73%) for PSA values < 0.5 ng/mL and 88% (95% CI: 77–96%) for PSA values ≥ 0.5 ng/mL, and the difference among these subgroups was statistically significant [12]. Similar findings have also been observed by Calais J et al. while evaluating the BCR of PCa with 68Ga-PSMA-11PET CT [13]. In our study also, 18F PSMA PET CT was able to detect tracer avid lesions even at very low PSA levels; a tracer avid lesion could be seen in the indeterminate group at low PSA levels of 0.14 ng/mL and a definite positive lesion could be found on PSA levels as low as 0.18 ng/mL. The high detection rates even at very low PSA levels seen in our study are in sync with the available literature.

Further, in our analysis, a PSA value of even ≥1 ng/mL was not found to be statistically significant for finding a tracer avid lesion in patients with BCR. We attribute this observation as a true reflection of real-world scenario where a mixture of patient populations with various T, N stages are referred and the PSA levels might not be the most recent. Nevertheless, we still could draw some inferences of practical importance from our analysis; we observed that following radical prostatectomy, a PSA of less than 0.08 ng/mL would result in a negative scan and a negative PSMA PET CT could be seen up to PSA 3.5 ng/mL while evaluating biochemical recurrence. Similarly indeterminate findings could be seen in patients with PSA levels of up to 6.5 ng/mL while evaluating BCR, which would most commonly be seen in lymph nodes. In patients having prostate gland, PSMA PET CT was positive if PSA level ≥ 0.71. Further, in this group, the majority would have a positive focus on the prostate gland either alone or in various combinations of the seminal vesicle, lymph nodal and systemic involvements.

In our study, the most common positive findings were seen in lymph nodes with pelvic lymph nodes being most commonly involved with the rest of the structural involvement being less in other combinations. Similar observations have been seen in previous studies [14,15,16]. Giesel FL et al. while evaluating lymph nodal recurrence of prostate cancer found the mean volume of lymph nodes was 0.5 mL. In 33% of their study population, at least one node was larger than conventional criteria for morphological positivity. In their study, 36% of patients had lymph nodes ≥ 8 mm and in patients with PSMA-positive lymph nodes (67%), none of the PSMA-positive lymph nodes met the morphological criteria for positivity. In their study due to subcm, PSMA avid lymph nodes, the N stage was changed from N0 to N1 in 67% of the cohort on ^68^Ga-PSMA ligand PET CT [14]. Rahbar K et al. [15], in their study, while evaluating diagnostic performance of ^18^F-PSMA-1007 in patients with biochemical recurrent prostate cancer, found the patient-based sensitivity to be 95%. In their analysis, the maximum number of recurrences was also found in lymph nodes. They found a total of 213 lesions characteristic of PCa of which 37 were local relapses, 107 lymph node metastases, 67 bone metastases and two soft tissue metastases. In 29 patients, relapse was seen exclusively in lymph node metastases. Sprute K et al., in their study of the staging of prostate carcinoma in primary and biochemical recurrence, found that in 34.4% of the cohort, positive lymph nodes were present on imaging. In their study, the patient-based analysis showed a sensitivity of 85.9% and a specificity of 99.5% for lymph nodes larger than 3 mm [16]. We did not evaluate further the pattern of distant spread, correlation with pathology, PSA dynamics, and levels in our current study as these have been studied in earlier studies with few showing their significance; furthermore, this was not the aim of our study.

In our study, we found that approximately one-third of the time (31.5%) a tracer avid lesion was seen in the prostate gland in the case of BCR. Multiple other studies performed for biochemical recurrences have suggested local recurrence from 23% to as high as more than 65% involving various patient populations. Ahmadi et al., while evaluating biochemical recurrence in a mixed population post brachytherapy, high-intensity focused ultrasound (HIFU) and RP, found an 80% positivity rate. In their study group, 23% showed local recurrence, 43% showed involvement of lymph nodes and 33%, 11% showed bone, soft tissue lesions. No multivariable model could be constructed for predictors of overall scan positivity; their findings were in concordance with our observations [17]. Mingels, C et al., while evaluating biochemical recurrence, reported a high positivity rate of 91% and also observed increased PET positivity with rising PSA levels similar to our study. On a regional basis in their study, PSMA avid lesions were detected in the prostatic fossa in 51% of the population, in pelvic LN in 48%, in retroperitoneal LN in 23%, in supradiaphragmatic LN in 16%, in bones in 53%, and in other metastasis (soft tissue lesions) in 7% of the population [18].

Further, in the non-prostatectomy group with biochemical recurrence, 75% of scans showed definite recurrence involving the prostate. A few studies focusing on BCR in post-radiotherapy settings have also reported similar high local recurrence detection rates ranging from 48 to about 64% following radiation therapy [19,20,21]. Hruby et al., in their study, observed that 31% of men had recurrence within the gland, 11% in the lymph node, 3% in bone and 1% in both. On further analysis, they found 17% of the population had an isolated located recurrence and, interestingly, it represented 2% of patients who received definitive EBRT and were followed for at least 2 years [19]. Einspieler I et al., in their study of BCR evaluation using hybrid 68 Ga PSMA PET CT in post-radiotherapy patients found that approximately 91% of patients showed tumor recurrence, with a higher positivity rate on higher PSA levels in this population. They found that approximately 64% of the population had local recurrence, distant lesions were seen in approximately 60% population while approximately one-fourth of the population in their study group had both local recurrence as well as distant lesions [20].

Unfortunately, the high local recurrences have not yet translated into directed interventions towards reducing local recurrences. We feel that it should be evaluated and discussed in further studies. A higher recurrence in the prostate bed while evaluating biochemical recurrence provokes us to ask if prostatectomy should be offered more proactively.

Further, our study reflects the importance of follow-up PSMA PET CT for the characterization of indeterminate findings of PSMA PET CT. In follow-up, findings turned definite in 22/30 patients (73.3%) with 14 patients having definite positive uptake (PSA rise 0.7–3.0 ng/mL in 6 months) and eight patients having definite negative uptake (PSA rise 0.13–2 ng/mL in 6 months) in prior indeterminate lesions. The PSA velocity was not considered statistically significant in our study, and this could be due to the small number of this group in our study population. Our study findings of a few patients with PSMA PET CT scans turning positive later on with further rise of PSA levels can be explained based on observations of Jia J et al. and Bashir et al. In their study, Jia J et al. found negative 18 F PSMA PET CT results in 22/114 patients. Seven of these received ADT before and after the 18 F PSMA PET CT scan. Fifteen patients were followed on PSA value and did not receive any treatment. The patients, who were treated with ADT, showed an early response to PSA; but PSA levels further increased in patients who were followed up without receiving any treatment [21].

Similarly, Bashir et al. found 11 of 28 BCR patients had negative ^18^F-PSMA PET CT results after the initial PCa surgery. Three patients in their study group received prostate bed salvage radiotherapy, and eight patients did not receive any treatment and were only followed up on PSA. The PSA level of all patients receiving treatment decreased, and the PSA level of follow-up patients increased. From their studies, it can be understood that a negative PSMA PET CT in cases of BCR following radical treatment can lead to an underestimation of local recurrence. This is further proven by the fact that few patients who received further treatment despite negative PSMA PET CT had a reduction in PSA levels on follow-up [22]. Similar significances of a negative PSMA-PET/CT have been suggested by Emmett et al. while evaluating treatment outcomes of salvage radiation treatment in men with rising PSA after radical prostatectomy; their study showed high PSA-response rates of about 85% to salvage radiotherapy in patients who received stereotactic radiotherapy to the pelvis despite negative PSMA-PET CT, suggesting pelvis-confined disease in the majority of patients with a negative PSMA-PET/CT, especially within the prostatectomy bed. They further commented that a false negative PSMA PET CT could be due to small lesions getting obscured in very close proximity to high urinary PSMA activity. In our study, urinary activity was not considered a significant factor due to the use of the 18 F labelled PSMA tracer which is considered superior in localizing local disease [23].

In 30 patients who had repeat PSMA PET CT to monitor small volume positive disease, 24 patients had new lesions with four patients having new uptake in the prostate (average PSA rise 2.1), four having new lesions in the prostate bed (average PSA rise 3), 12 having new pelvic lymph node (average PSA rise 3.8) and four having new bone lesions (average PSA rise five). Eight patients had negative PSMA PET CT on repeat imaging also (PSA rise 0.3–2.3). Although a higher PSA rise was seen in patients with new PSMA avid lesions, we were unable to draw a statistical significance of this small number.

Our present study has a few limitations. One of the limitations is the retrospective nature of analysis and another is the lack of histological proof of our scan findings as most of the findings, particularly the lymph nodes, could not be histologically confirmed. Unfortunately, this problem is commonly observed in radiological studies evaluating BCR. As most of the lesions detected on scans are located deep in the pelvis and are not easily accessible, pathological confirmation of most of the scan findings is difficult. However, due to the use of standardized PSMA RADS criteria, we seem to have at least partially overcome this limitation as compared to earlier studies evaluating BCR. PSMA RADS criteria, reflect the likelihood of the presence of PCa; PSMA RADS-1, 2 lesions are either certainly or almost certainly benign, PSMA-RADS-4 indicates a high likelihood PCa presence and PSMA-RADS-5 lesions almost certainly represent PCa, we seem to have achieved a fair amount of certainty towards our findings.

Overall, our findings are significant as they reflect a real-world scenario of care in a tertiary care centre and we believe our findings can lead to definite changes in management and referral criteria. We believe further that prospective studies would be helpful.

## 5. Conclusions

Our study reveals a notably higher recurrence rate in the prostate bed while evaluating biochemical recurrence, with approximately one-third of cases showing involvement of the prostate gland. Interestingly, the prevalence of prostate gland involvement is significantly higher in the non-radical prostatectomy group, comprising 75% of the study population with biochemical recurrence.

Furthermore, we found that conducting a follow-up PSMA PET CT scan can yield decisive results in cases with prior indeterminate findings. Specifically, in this group, a PSA rise of 0.7 ng/mL within a six-month period tends to lead to definitive positive findings on the follow-up scan. On the other hand, it is noteworthy that a follow-up scan can still yield negative results even after a PSA rise of 2 ng/mL within the same six-month interval. These valuable insights can be of great assistance to referring clinicians in managing patients with suspected biochemical recurrence.

## Figures and Tables

**Figure 1 diagnostics-13-02542-f001:**
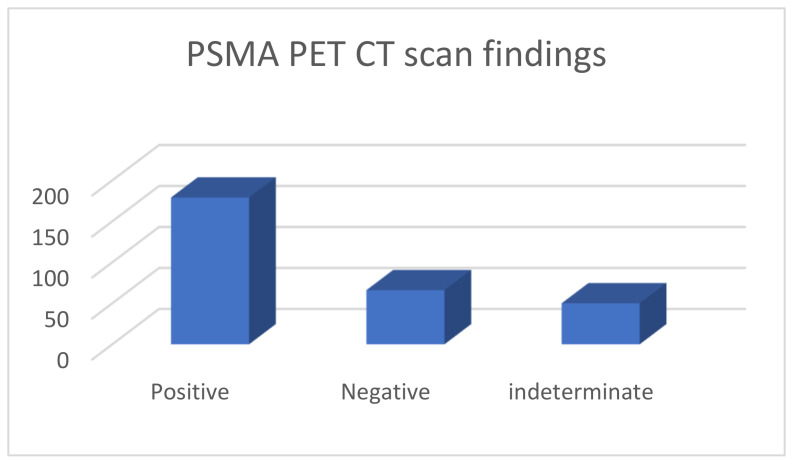
PSMA PET CT scan findings.

**Figure 2 diagnostics-13-02542-f002:**
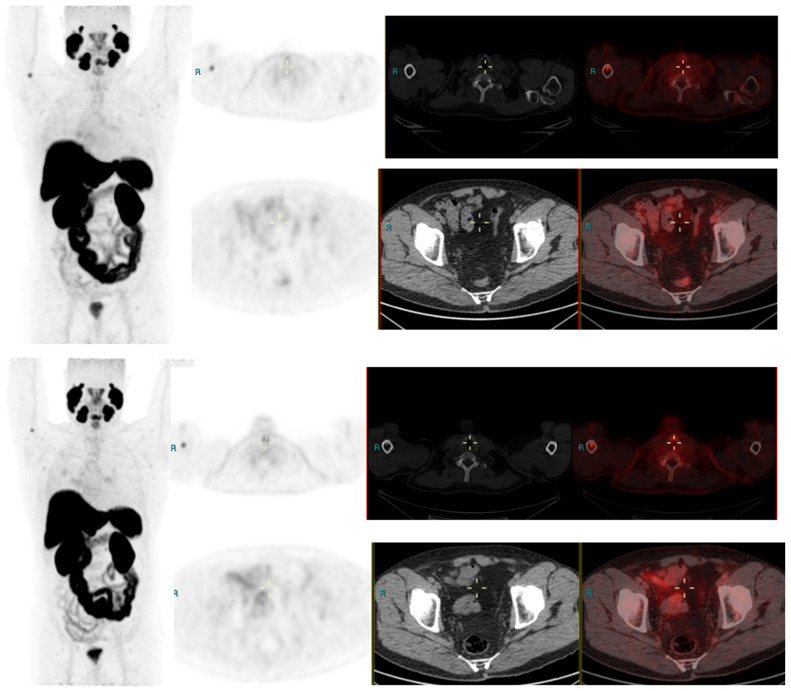
Radical prostatectomy—Negative scan at PSA 2.1 and 3.4 ng/mL, PSA rise of 1.3 ng/mL after 6 months. PSMA RADS 1 B, due to uptake in prior known humeral fracture.

**Figure 3 diagnostics-13-02542-f003:**
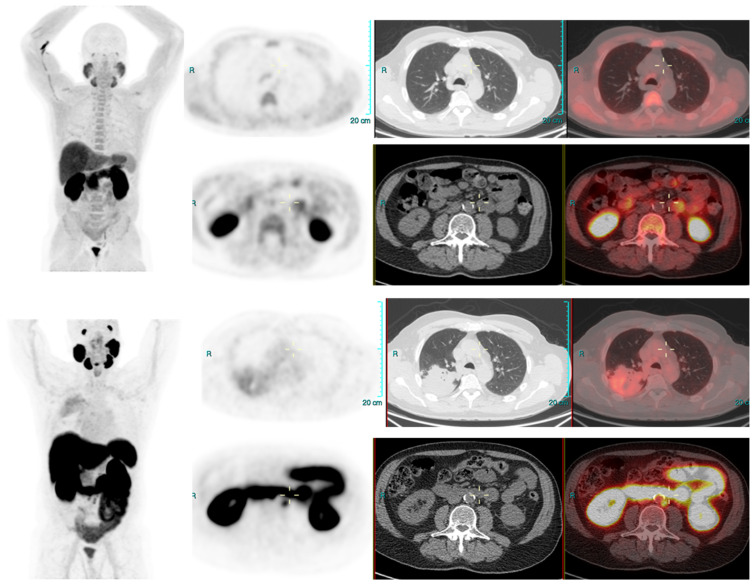
Radical prostatectomy negative scan at PSA 3.5 ng/mL, indeterminate lymph node at PSA 6.5, rise of 3 ng/mL in 2 years RADS 3A—RP status. Consolidation in right upper lobe. Resolved on follow up CT after 2 months.

**Figure 4 diagnostics-13-02542-f004:**
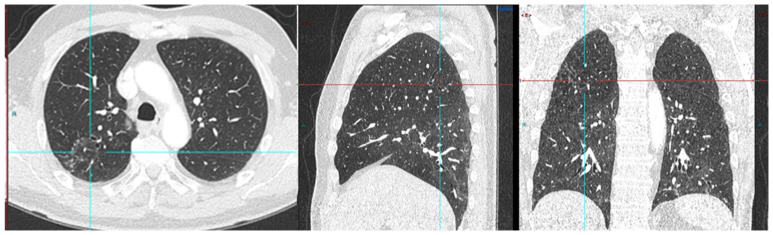
PSA 0.9 ng/mL. Positive lymph node metastasis and suspicious uptake in prostate.

**Table 1 diagnostics-13-02542-t001:** Patient characteristics of study.

Patient Characteristics.			
	Positive	Negative	Indeterminate
Age (range) in years	47–92	49–84	47–91
PSA range ng/mL	0.18–99.7 ng/mL	0.08–3.5 ng/mL	0.14–6.5 ng/mL
Mean PSA	8.62 ng/mL	2.5 ng/mL	3.0 ng/mL
PSA ≥ 0.5 ng/mL	145	18	33
PSA < 0.5 ng/mL	34	48	33
PSA ≥ 1	132	26	19
PSA < 1	47	24	47
Most common stage	T3 (51.3%)	T3 (50%)	T3 (52%)
T1	13	8	7
T2	36	12	10
T3a	31	22	16
T3b	61	11	10
T4	38	13	7
Most common GS	7 (4 + 3)	7 (3 + 4)	7 (4 + 3)
Radical Prostatectomy	67 (37.3)	61 (92.4%)	28 (56%)
Radiotherapy (Prostate+/−LN)	112		22
Salvage Radiotherapy after RP		22	12

**Table 2 diagnostics-13-02542-t002:** Positive group features.

PSMA PET CT Positive Group			
Radical Prostatectomy (RP)	66	Non-RP	112
Pelvic L.N	33	Only prostate	25
Pelvic L.N and bones	13	Prostate bed	17
RP L.N and bone	13	Prostate and extra prostate	42
Prostate bed and pelvic l.n	4	Pelvic LN	2
Bones	2	Pelvic and RP LN	6
Bone with soft tissue	1	Systemic (bone + visceral)	20

**Table 3 diagnostics-13-02542-t003:** PSA cutoff for PSMA avid and non avid lesions.

	PSMA Avid Lesion (Positive/Indeterminate)	Negative Scan
PSA ≥ 0.5	145 + 39 = 184	33
PSA < 0.5	34 + 11 = 45	33
	229	66
PSA ≥ 1	132 + 26 = 158	19
PSA < 1	47 + 24 = 71	47
	229	66

**Table 4 diagnostics-13-02542-t004:** PSA velocity among patients.

Follow up Imaging PSMA PET CT	61	PSA Rise in 6 Months
Indeterminate findings	30	
Positive	14	0.7–3
Negative	8	0.13–2
Same indeterminate focus	8	0.4–2.5
Repeat despite earlier definite positive/negative	31	
New lesion	24	0.3–10
Negative repeat	7	0.3–2.3

## Data Availability

Data is unavailable due to privacy or ethical restrictions.

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
