# Peer review of "Lessons from a 3-Year Review of PSMA PET-CT in a Tertiary Setting: Can We Fine Tune Referral Criteria by Identifying Factors Predicting Positivity and Negativity?"

_diagnostics, 2023, doi:10.3390/diagnostics13152542_

Round 1
Reviewer 1 Report
These authors have retrospectively analyzed PSMA-PET/CT scans which were performed following evidence for biochemical recurrence. The database which was used for this comprehensive analysis about the role of PSMA imaging in this setting included a relatively large number of scans (295) which spanned over 3 years. The data generated from this analysis clearly demonstrates the role of PSMA-PET/CT imaging in assessing patients with suspected recurrent disease at the primary sites as well as other locations throughout the body.
Overall, the data are very solid and shows the unavoidable applications of this approach in the management of this common and somewhat aggressive cancer.
However, the authors emphasize the importance of PSMA imaging in detecting lymph node metastasis as being a major contribution of this technique in this setting. We have to point out that the sensitivity of the PET/CT imaging in detecting lymph node metastasis in general is suboptimal and lesions that are outside the spatial resolution of this modality will be frequently be missed on PSMA scans. While phantom studies indicate that this imaging instrument has a spatial resolution of several millimeters, in reality, it is more like 8-10 mm and therefore this gross imaging technique will be unable to detect small lesions with high sensitivity. Therefore, the majority of microscopic lesions will be missed by PET imaging which will adversely impact optimal management of these patients. The suboptimal performance of PET has been described in the following scientific communication: Alavi A, Carlin SD, Werner TJ, Al-Zaghal A. Suboptimal Sensitivity and Specificity of PET and Other Gross Imaging Techniques in Assessing Lymph Node Metastasis. Mol Imaging Biol. 2019 Oct;21(5):808-811. doi: 10.1007/s11307-018-01311-4.
Author Response
Dear Reviewer,
Thank you for raising this point.
I would like to bring your attention to a clarification regarding the size of lesions mentioned in our scientific communication. The statement "Overall, the spatial resolution of conventional PET body scanners is approximately 8-10 mm" pertains to conventional PET scanners as stated in the referenced study. However, it is important to note that in our study, PET acquisition was performed in the time-of-flight mode, which has been shown to overcome some of the limitations in spatial resolution mentioned in the literature and this has been mentioned in the referenced study also.
Furthermore, I would like to highlight the role of PSMA PET CT in the detection of subcentimeter lymph node metastases, as discussed in the review article by Matteo Ferrari and Giorgio Treglia titled "18F-PSMA-1007 PET in Biochemical Recurrent Prostate Cancer: An Updated Meta-Analysis," which is referenced as number 5 in our bibliography. The review article states that regional and distant lymph nodal metastases, even with a short-axis diameter of less than 1 cm, as well as local relapse and bone metastases, were frequently detected using F-PSMA-1007 PET/CT.
In our study, we also observed that the majority of lymph nodes detected were sub centimetre size. This can be attributed to the fact that all the patients in our study had previously undergone evaluation using other cross-sectional imaging modalities such as MRI/CT for biochemical recurrence prior to the PSMA PET CT scan. Although we had mentioned this in our manuscript, we have now restructured the Material and Methods section to provide clearer emphasis on this point in the revised manuscript.
Reviewer 2 Report
This was an Audit of PSMA PET results and not a prospective trial or even a case-control study.
Often positive appearances of Prostate Cancer Recurrence within the Prostate following Radiotherapy are false positive and these findings need to confirm with a biopsy.
The numbers of positive scans are small for them to be then stratified into groups. There is no statistical test to correlate sites of relapse with treatment.
The following comment in Discussion is not supported by your findings and doesn't correlate with the current evidence. Not sure how performing more prostatectomies will improve outcomes?
'Unfortunately, the high local recurrence have not yet translated into directed interventions towards reducing local recurrences. We feel that it should be evaluated and discussed in further studies. A higher recurrence in prostate bed while evaluating biochemical recurrence provokes us to think if prostatectomy be offered more proactively?'
Author Response
- This was an Audit of PSMA PET results, and not a prospective trial or even a case-control study.
Clarification 1:
Thank you for raising this important point. We can confirm that this project has been labelled as a “Service Evaluation” and hence conforms to all appropriate related governance issues.
- Often positive appearances of Prostate Cancer Recurrence within the Prostate following Radiotherapy are false positive and these findings need to confirm with a biopsy.
Clarification 2:
Thank you for raising this query. We have actually acknowledged this fact in the Discussion part (page 9, para 3), where we have stated ‘Unfortunately, this problem is commonly observed in radiological studies evaluating BCR. As most of the lesions detected on scans are located deep in the pelvis and are not easily accessible, pathological confirmation of most of the scan findings is difficult. However, due to use of standardized PSMA RADS criteria, we seem to have at least partially overcome this limitation as compared to earlier studies evaluating BCR. We seem to have achieved a fair amount of certainty towards our findings.’
In our study, approximately one-third of BCR cases exhibited tracer avid lesions in the prostate gland, with a notably higher prevalence in the non-prostatectomy group, where 75% of scans showed definite prostate recurrence. These findings align with multiple studies on biochemical recurrences, reporting local recurrence rates ranging from 23% to over 65% across various patient populations. Studies focusing on BCR in the post-radiotherapy setting have similarly reported high local recurrence detection rates, ranging from 48% to about 64% following radiation therapy (References 17-21).
- The numbers of positive scans are small for them to be then stratified into groups. There is no statistical test to correlate sites of relapse with treatment.
Clarification 3:
Indeed, the statement is true. Given the small subgroups in our study, determining statistical significance of our findings becomes challenging. However, our study's results can serve as a foundation for addressing these questions in future prospective studies with larger patient populations. This is a study capturing “Real-World Experience” of using PSMA scans, as mentioned in the title and introduction. Our descriptive analysis is therefore guided by the cohort of patients that were referred and underwent scans.
- The following comment in Discussion is not supported by your findings and doesn't correlate with the current evidence. Not sure how performing more prostatectomies will improve outcomes? 'Unfortunately, the high local recurrence have not yet translated into directed interventions towards reducing local recurrences. We feel that it should be evaluated and discussed in further studies. A higher recurrence in prostate bed while evaluating biochemical recurrence provokes us to think if prostatectomy be offered more proactively?'
Clarification 4: Thank you for raising this fundamental question for practice of medicine. Our findings suggest that in patients with BCR, it is common to find recurrence within the prostate itself. Similar findings have also been encountered in prior studies, and as we have stated, unfortunately, it has not been translated into interventions to reduce local recurrences. As, these patients have already received radiotherapy earlier, it makes us wonder and raise the question that the parameters and decision-making prior to radical prostatectomy should perhaps be revisited. Perhaps the findings of studies such as this will re-focus efforts on reducing local recurrence in the prostate bed, and it is possible that some patients will opt for radical prostatectomy. While we are not advocating for more radical prostatectomies, it is possible that patients may choose this option once they are made aware of the possible strategies to manage biochemical recurrence.
Round 2
Reviewer 2 Report
Even though some improvement in describing methodology I don't agree with the conclusion.
The findings do not point towards Prostatectomy being superior to other treatment modalities.
Also study underpowered to draw any conclusions
Author Response
- Thank you for your comment and appreciation.
- We have removed this point from abstract as well as our conclusion.
- Our findings are as conclusive as can be in the real world settings and probably future prospective studies will be able to draw more conclusive data.
Round 3
Reviewer 2 Report
Accepted corrections